

# A new tool for the estimation of Ground-Based InSAR acquisition characteristics before starting installation and monitoring survey

Charlotte Wolff[1], Marc-Henri Derron[1], Carlo Rivolta[2], Michel Jaboyedoff[1]

[1]ISTE, University of Lausanne, Lausanne, 1015, Switzerland
[2] Ellegi srl, Milano, 20123, Italy

*Correspondence to*: Charlotte Wolff (charlotte.wolff@unil.ch)

**Abstract.** Synthetic Aperture Radar (SAR) acquisition can be performed from satellites or from the ground by means of a so-called GB-InSAR (Ground-Based Interferometry SAR), but the signal emission and the output image geometry slightly differ between the two acquisition modes. Those differences are rarely mentioned in the literature. This paper proposes to compare

satellite and GB-InSAR in terms of (1) acquisition characteristics and parameters to consider; (2) SAR image resolution; (3) geometric distortions that are foreshortening, layover and shadowing.

If in the case of satellites SAR, the range and azimuth resolutions are known and constant along the orbit path, in the case of GB-InSAR their values are terrain-dependent. It is worth estimating the results of a GB-InSAR acquisition one can expect in terms of range and azimuth resolution, Line of Sight (LoS) distance and geometric distortions to select the best installation

location when several are possible. We developed a tool which estimates those parameters from a Digital Elevation Model (DEM), knowing the GB-InSAR and the Slope of Interest (SoI) coordinates. This tool, written in MATLAB, was tested on a simple synthetic point cloud representing a cliff with a progressive slope angle to highlight the influence of the SoI geometry on the acquisition characteristics and on two real cases; cliffs located in Switzerland, one in the Ticino canton and on in the Vaud canton.

# 1    Introduction

The use of Synthetic Aperture Radar Interferometry (InSAR) as a remote sensing technique capable of detecting and monitoring small ground displacements started in the 80s (Gabriel et al. 1989). It was firstly used specifically with space borne platforms such as satellites ERS-1 (1991) or RADAR-SAT (1995, Zebker et Villasenor 1992; Massonnet et al. 1993; Usai et Hanssen 1997). In the field of geosciences, it was dedicated to studying small movements phenomena (less than cm) at a

decametric to metric resolution over large areas (km), such as subsidence (Cabral-Cano et al., 2008; Strozzi et al., 2018), volcanic activities (Wicks et al. 1998; Garthwaite et al. 2019) or landslides (Tarchi 2003; Hilley et al. 2004; Colesanti et Wasowski 2006). Since then, the technique expanded and in the late 90s first radar devices monitoring displacements from a Ground Base (GB) were deployed (Tarchi et al., 1997; Cazzanil et al., 2000; Pieraccini and Miccinesi, 2019). Some use a Real Aperture Radar (RAR) antenna (Werner et al., 2008) when others consider a Synthetic Aperture Radar (SAR, Rudolf et al.,

1999; Leva et al., 2003; Antonello et al., 2003).



Satellite- and GB-InSAR are complementary, both detecting displacements only along their respective Line-of-Sight (LoS) (Casagli et al. 2003; Catani et al. 2014; Carlà et al. 2019). InSAR satellites detect sub-vertical movements whereas GB-InSAR gather information on sub-horizontal movements (Wolff et al., 2022).

Radar images acquisition and processing are sensitive to the terrain geometry which can have significant effects on the appearance of the resulting radar image such as slope compression and highlighting (foreshortening and layover effects, (Jensen, 2006) or surfaces not illuminated by the radar appearing dark and elongated in the image (shadowing effect). Terrain slopes and radar incidence angle influence the image resolution. Satellite imagery-oriented software creates foreshortening and layover masks (Kropatsch and Strobl, 1990; Rees, 2000). While SAR satellite geometries are well documented (Griffiths 1995; Rees 2000; McCandLOSess et Jackson 2004; Ferretti et al. 2007), the transposition of these geometries in the case of 40 GB-InSAR is seldom mentioned in the literature. Nevertheless, when starting a new GB-InSAR campaign, it is worth estimating what results one can expect in terms of distance to the region of interest, range, and azimuthal resolutions as well as potential foreshortening and shadowing effects; in order to select the best position before starting the campaign. This can specifically be the case when installing a GB-InSAR in remote areas and difficult installation sites (Lingua et al. 2008; Cafduff et al. 2015; Talich 2016; Rouyet et al. 2017).

After transposing the SAR geometry described for satellites to ground SAR geometry and presenting the main differences between satellite and GB-InSAR, this paper describes a MATLAB tool with a user interface, designed to compute several parameters of the radar image such as its range and azimuthal resolutions, the areas affected by shadowing or strong foreshortening in the case of a Linear SAR system consisting in a radar measuring head translating along a rail. The needed inputs are a Digital Elevation Model (DEM) in an ascii format, the localization of the area of interest and the localization where 50 one intends to install the GB-InSAR. The main objective is to provide a tool for helping surveyors to find the best installation location.

The tool has been tested on three study cases: (1) a synthetic cliff made of slope angles increasing form the bottom to the top, (2 & 3) two real instable cliffs that have been monitored with a GB-InSAR, Cima del Simano and La Cornalle. For Cima del Simano, the results for three different radar positions were compared to select the best installation position.

## 2 Theory

**Table 1 : List of abbrevations used for the radar caracteristics presented here after.**

| Name | Abreviation | Unit | Definition |
|---|---|---|---|
| Line-of-Sight | $\overrightarrow{LoS}$ | Vector | Radar to target direction |
| LoS distance | dLoS | m | Radar-target distance |
| Distance on slope | $D_{slope}$ | m | Distance of illuminated surface along the slope, in range direction |
| Look or nadir angle | $\Phi$ | ° | Angle between the vertical line and the LOS |





| Incident angle | θ | ° | Angle between LoS and the normal of the targeted surface |
|---|---|---|---|
| Depression angle | γ | ° | Complement angle to Φ |
| Apparent Orientation | ω | ° | Horizontal angle between LoS and target slope strike |
| Slope dip | α | ° | Slope dip |
| Apparent slope dip | $\alpha_{app}$ | ° | Apparent slope dip seen from radar position |
| Speed of light | c | m.s$^{-1}$ | $3.10^8$ |
| Radar wavelength | ʎ | cm | Spatial period of the signal, varying 0.8 cm and 10.0 cm for radar |
| Frequency | f | GHz | f=c/ʎ |
| Pulse length | τ | µs | - In the case of satellite InSAR, duration of the emission of one radar pulse<br>- In the case of GB-InSAR, duration of one sequence of frequency variation |
| Pulse repetion interval | PRI | µs | - In the case of satellite InSAR, time between the emission of two consecutive radar pulses |
| Pulse repetition Frequency | PRF | MHz | PRF = 1/ τ |
| Frequency Bandwidth | BW | MHz | In the case of a continues signal (GB-InSAR), difference between the upper and lower cut-off frequencies<br>BW = 1/ τ (Mahafza, 2000) |
| Antenna Beamwidth | ε | ° | Angle from which the majority of the antenna's power radiates<br>In the case of GB-InSAR, the vertical and horizontal beamwidths are different and denoted $\varepsilon_v$ and $\varepsilon_h$ |
| Illuminated area length | $W_{illu}$ | m | Length of the illuminated area in the case of the GB-InSAR, which increases with the range. |
| Illuminated area height | $H_{illu}$ | m | Height of the illuminated area in the case of the GB-InSAR |
| Synthetic antenna length | L | m | - In the case of Linear GB-InSAR, rail length used to focus the radar image (which is shorter than the total rail length)<br>- In the case of satellite InSAR, L can be infinite |
| Real antenna length | $L_{real}$ | | - In the case of satellite InSAR, radar antenna length |
| Resolution | R | m | Size of the smallest object detectable by the sensor |
| Ground range resolution | $R_r$ | m | Vertical resolution of the radar image |
| Azimuthal resolution | $R_{az}$ | m | Horizontal resolution of the radar image |





## 2.1 SAR Geometry

The geometrical characteristics of radar imagery differ from standard image geometry (Lin et Fuh 1998; Turner et al. 2021).
In the case of radar imagery, some parameters need to be defined and distinguished when applied to satellite and aerial InSAR or to GB-InSAR. The abbreviations are summarized in Table 1.

### 2.1.1 LoS, azimuthal and range directions

In the case of aerial radar, the azimuthal direction corresponds to the direction of the displacement of the aircraft or satellite; the range direction, or look direction is the direction perpendicular to the azimuthal direction; and the direction of the radar-
to-target line is the LoS direction whose distance is called range or LoS distance (dLoS). It varies from near-range, for the line forming the smaller angle with the vertical radar-Earth line (nadir), to the far range for the direction with the larger angle (Figure 1 a, b).

### 2.1.2 Look angle or off-nadir angle Φ, incident angle θ and depression angle γ

Those angles are well-defined in the case of satellite imagery when the surface monitored is assumed to be sub-horizontal. The
look angle Φ is the angle between the vertical line and the LoS. The depression angle γ is the complement angle of the look angle. The definition of the incident angle θ is the same as in optical geometry, i.e. the angle between the LoS and the normal to the monitored surface (Figure 2a, b). To simplify, when the monitored surface is horizontal for satellite InSAR or vertical for GB-InSAR, both θ and Φ are the angle between the LoS and the normal to the observed surface and are thus assumed to be equal.

### 2.1.3 Antenna beamwidth ε


The emitted signal propagates within a certain emission cone defined by an angle called beamwidth ε (Woodhouse, 2006; Miron, 2006) which is proportional to the wavelength according to diffraction laws (Lipson et al. 1995) and defines the maximum extent of the illuminated area. The radar footprint on the ground is an ellipsoid (Figure 1a, b). Radar manufacturers provide antenna emission characteristics that are displayed in the form of a polar diagram (Toomay and Hannen, 2004). In the
case of GB-InSAR, the vertical beamwidth $\varepsilon_v$ is limited to 30° to avoid interferences with other radar devices such as planes (Anon, 2017: « ETSI EN 300 440 v2.1.1 ») while the horizontal beamwidth $\varepsilon_v$ is not legislatively restricted (Figure 1d).

### 2.1.4 Radar bandwidth BW vs radar pulse length τ and pulse frequency F

One of the major differences between GB- and satellite InSAR is related to their emitted signal. In the case of satellites, the amplitude of the signal sent must be important to reach the Earth surface and to be backscattered with enough intensity to be
recorded by the radar receiver (Ferretti et al. 2014). However, the antennas are not able to continuously generate and send such



a high-peak power signal. To overcome this technical limitation, the radar signal is sent by pulses defined by a certain duration τ comprised between 10 and 100 μs depending on the satellite (Figure 1d).

The European Telecommunications Standards Institute (ETSI) defined some standards regarding the Short-Range Devices (SRD) emitting radio signals (Anon, 2017). The power and the frequencies of signals sent by terrestrial radar are limited to

not interfere with other devices emitting and receiving radio signals (Table 2).

GB-InSAR is considered as a Frequency-Modulated Continuous Wave radar (FMCW radar, Wolff 1998; Nadav 2003), the signal emitted is of lower intensity. Since the signal is continuously emitted, one is lacking the timing mark necessary to isolate the backscattered signals and discriminate the range. This is done instead by modulating the frequency sent by the transmitter (Figure 1e). The radar bandwidth (BW, Figure 1g) is the difference between the upper and lower cut-off frequencies. The

duration of one sequence of frequency variation is also called pulse length τ and is linked to the BW by the following equation (Mahafza, 2000):

$$BW = \frac{1}{\tau} \qquad (1)$$

**Table 2 : Principal restriction concerning GB-InSAR signal, defined by ETSI**

| Parameter | Unit | Maximum limit |
|---|---|---|
| **Frequency range** | GHz | 17.1 to 17.3 |
| **BW** | MHz | 200 |
| **Maximum power output** | dBm | 26 |

### 2.1.5    Real Antenna length $L_{real}$ and synthetic antenna length L

The range resolution is inversely proportional to the real antenna length $L_{real}$ but to increase this resolution, synthetic aperture antennas L are used in the case of SAR acquisition. For a GB-InSAR installed on a rail, the antenna length L corresponds to the rail length used to focus the radar image, which is in practice slightly shorter than the total rail length.



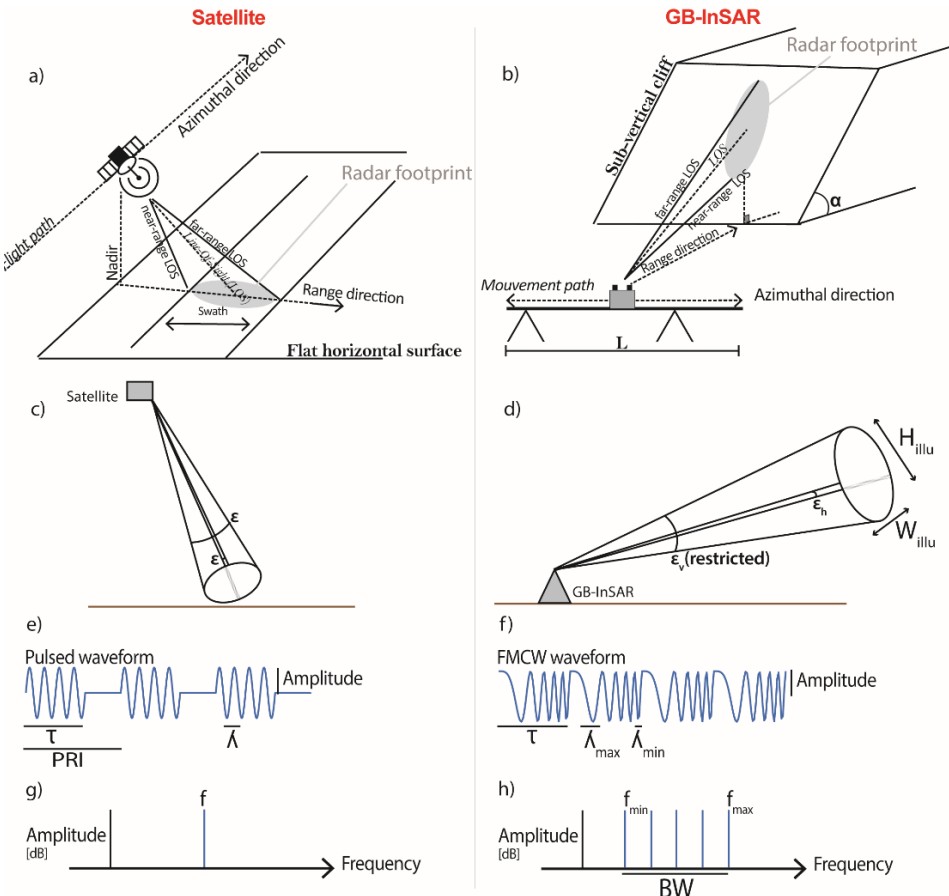

**Figure 1 : Illustration and comparison of radar acquisition characteristics in the case of satellite InSAR (Left) and GB-InSAR (Right). (a), (b) Geometry of the acquisition defining the azimuthal and range directions. (c), (d) Beamwidth characteristics. (e), (f) Waveform characteristics. (g), (h) Frequencies characteristics.**

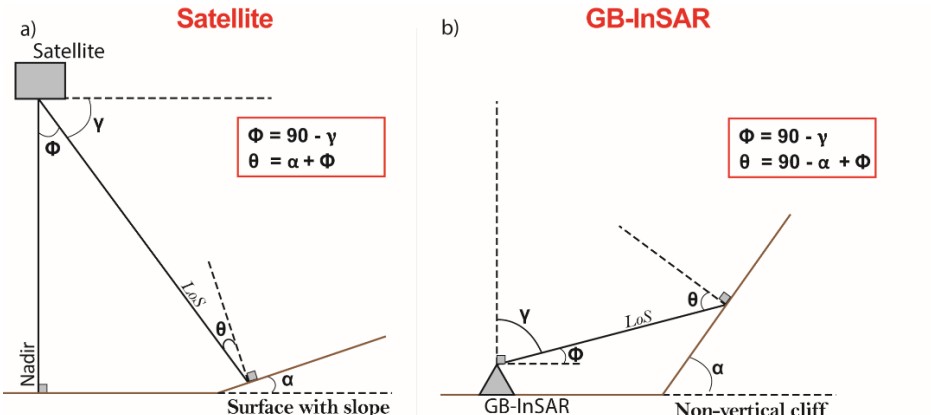

**Figure 2 : Illustration of the main specific angles used when describing a SAR acquisition. (a) Case of satellite radar acquisition. (b) Case of GB-InSAR acquisition.**





### 2.2 Spatial resolution

#### 2.2.1 Radar and optical images

When representing the world (Figure 3a) by an image, one must distinguish the radar image from the optical image, which is the visual display we are commonly used to (Figure 3b). The radar image is based on the distance between the radar antenna and each feature of the scene: the bottom line of the image corresponds to the monitored surface closest to the radar. Additionally, in radar geometry, range and azimuthal resolutions are defined differently and by default, pixels are not square (Figure 3c, d).

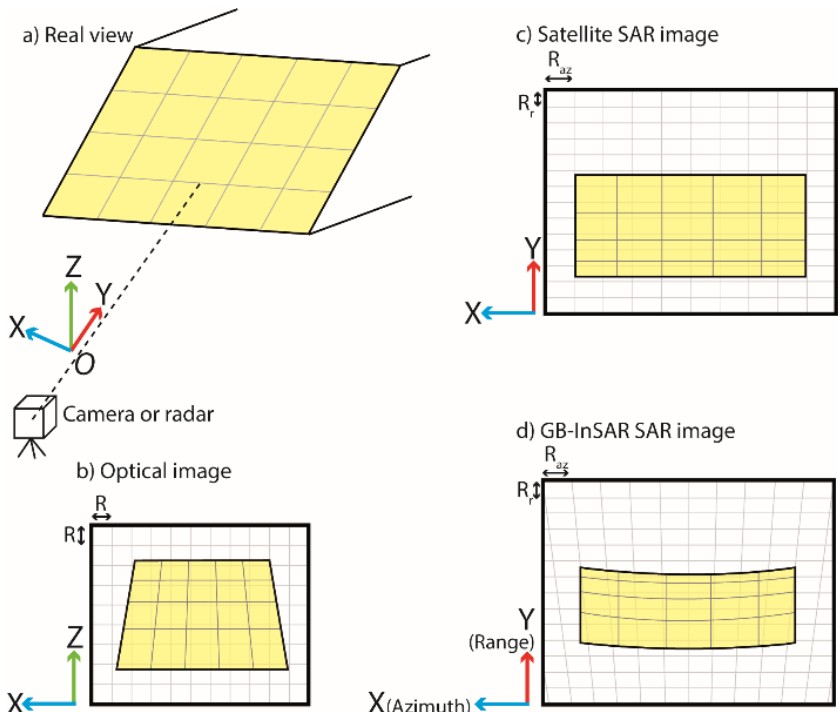

**Figure 3: Comparison of the view of a cliff. a) Real view in a parallel projection, b) Optical image taken with a camera, c) Satellite SAR image, d) GB-InSAR SAR image (after Tapete et al., 2013). In the case of the radar images, the slope is compressed due to a foreshortening effect and the distance between two horizontal lines increased along Y due to an increase of the range resolution. GB-InSAR footprint image is a cone, Y axis corresponding to the range distance dLoS, and the azimuthal resolution increases with the latter.**

#### 2.2.2 Azimuthal resolution $R_{az}$

The azimuthal resolution $R_{az}$ corresponds to the resolution parallel to the flying trajectory in the case of satellite InSAR or parallel to the rail in the case of a Linear GB-InSAR, or the horizontal resolution in the radar image. Its value differs between satellite InSAR and GB-InSAR.



The synthetic antenna aperture length being constrained by the rail length for the GB-InSAR, $R_{az}$ is related to the radar

beamwidth (itself related to the wavelength λ), the range distance dLoS and the synthetic antenna length L with the following

relation (Henderson and Lewis, 1998; Jensen, 2006):

$$R_{az,GB-InSAR} = \frac{dLoS\ \lambda}{2L} \qquad (2)$$

Thus, $R_{az}$ increases from near- to far-range and the GB-InSAR image footprint is a cone (Figure 3d). Conversely, for satellite

the synthetic aperture length L can be infinite, $R_{az}$ is dLOS-free and ʌ-free (Henderson and Lewis, 1998) and defined as:

$$R_{az,satellite} = \frac{L_{real}}{2}. \qquad (3)$$

**The satellite InSAR image footprint is thus rectangular (Figure 3c).**

Figure 4 presents the influence of dLoS and L on the azimuthal resolution in the case of the GB-InSAR.

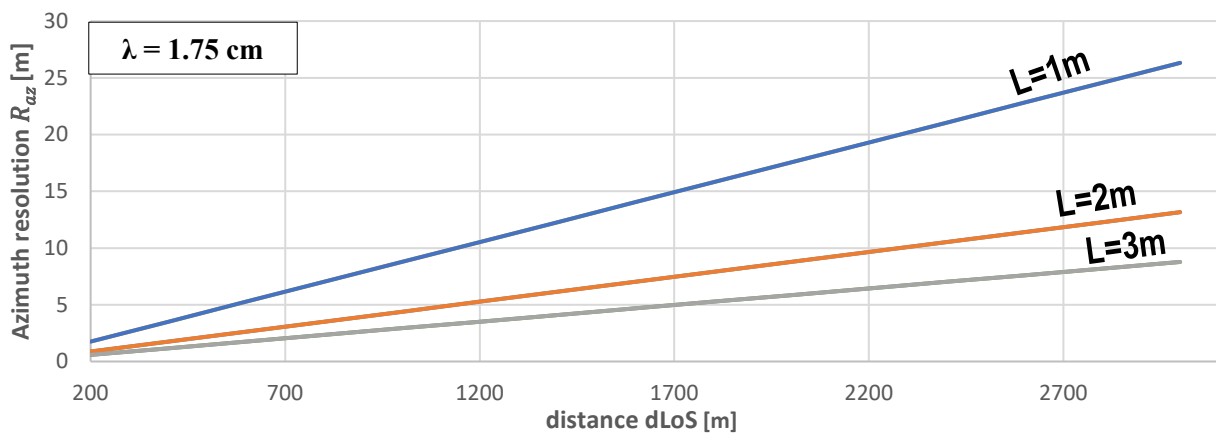


**Figure 4 : Influence of the rail length used to focus the GB-InSAR image corresponding to the synthetic antenna length L and the distance dLoS on the azimuthal resolution $R_{az}$, for a Ku-band with a wavelength equal to λ 1.75 cm. The longer the rail length, the better the azimuthal resolution.**

### 2.2.3    Ground range resolution $R_r$

The ground range resolution $R_r$ is the resolution along the LoS direction or the vertical resolution. It corresponds to the

minimum time needed to distinguish two consecutive pulses (Woodhouse, 2006). It is ground-geometry dependent, linked to

the incidence angle θ, the speed of light c and the pulse length τ according to the following relation (Henderson and Lewis,

1998; Jensen, 2006):

$$R_{r,satellite} = \frac{\tau.\ c}{2\sin\theta} = \frac{c}{2\ BW\sin\theta} = \frac{c}{2\ BW\sin(\Phi - \alpha_{app})} \qquad (4)$$


$$R_{r,GB-InSAR} = \frac{\tau.\ c}{2\sin\theta} = \frac{c}{2\ BW.\sin\theta} = \frac{c}{2\ BW\sin(90 - \alpha_{app} + \Phi)} = \frac{c}{2\ BW\cos(\Phi - \alpha_{app})} \qquad (5)$$



The monitored cliff geometry has an impact on the range resolution with two major consequences according to Equation 5: (1) Near-range surfaces and features possess, along a planar topography, less resolution in range than those in the far-range because Φ increases with the range and (2) steeper slopes increase the range resolution by increasing $\alpha_{app}$ (Figure 5, Sabins 1997; Stimson 1998; Jensen 2006). Furthermore, the shorter the pulse length τ, the finer the resolution. Nevertheless, one must

be careful in the choice of the pulse length because if a short pulse length results in a better resolution, the backscatter signal is also weaker and might not be detected if it is too low. For GB-InSAR, the parameter that can be chosen by the user is the bandwidth BW, since τ and BW are linked by Equation 1. The further away the radar is installed from the target area, the smaller should be the bandwidth to be sure to detect the backscattered signal. A good balance between an acceptable resolution and backscattered enough signal must be found (Figure 6a, b).


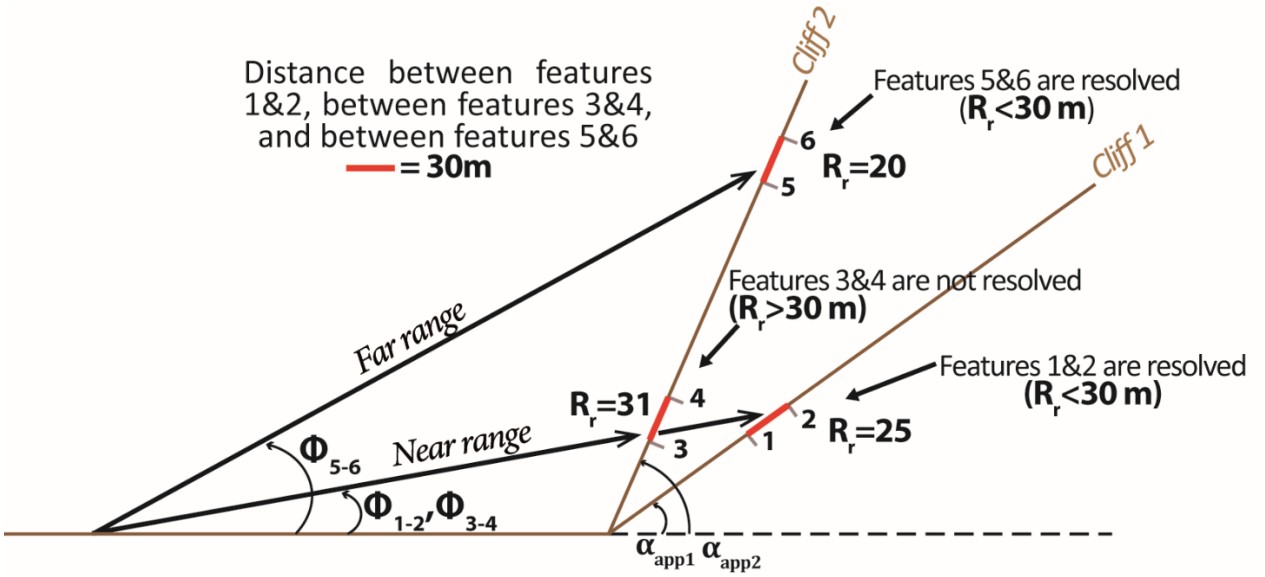

**Figure 5 : Ground range resolution Rr  for two different slope angles and two different depression angles (adapted from Sabins 1997). The distance between features 1&2, features 3&4 and features 5&6 is the same, 30 cm. However, features 1 and 2, located on a gentle slope and at a near-range distance from the GB-InSAR, are resolved (Rr = 25 m) as well as features 5 and 6 located at a far-**
**range distance and on a steep slope (Rr = 20 m), while features 3 and 4 located on the same steep slope as features 5 and 6 but at a near-range distance, are not resolved (Rr = 31 m).**





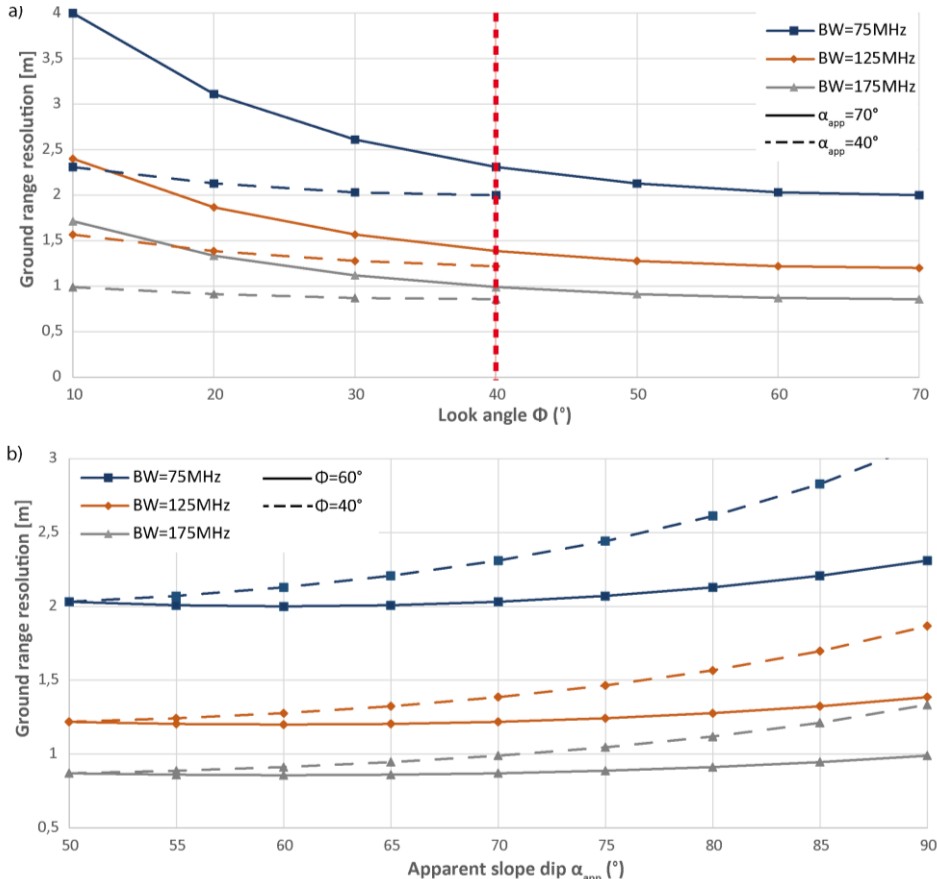

**Figure 6: Influence of some radar parameters on the range resolution R_r. a) Influence of the look angle Φ for different BWs. It is**
**interesting to notice that the resolution varies little when Φ is larger than 40°. b) Influence of the apparent slope dip α_app for different**
**BWs.**

### 2.2.4    Number of pixels and slope compression

During a monitoring campaign, the surveyor focuses on a Slope of Interest (SoI). A good monitoring is when the information

related to this SoI is distributed in the maximum of pixels in range and not compressed only in a few ones; this corresponds to

trying to reach the smaller range resolution. The position of the GB-InSAR will have an influence on the resolution and the

number of pixels in which the SoI will be contained can be estimated with the formula:

$$Nb_{pixelRange} = \frac{D_{SoI}}{R_r} \qquad (6)$$

where $D_{SoI}$ is the distance along the monitored slope of the SoI.

The steeper the slope, the more the information will be compressed, and some interesting features may be contained in the

same pixel. Given this consideration, if one has the choice between two radar installations, it is worth (1) reducing the range

distance (Figure 7a) and (2) reducing the apparent slope angle of the measured cliff to increase the apparent SoI distance

(Figure 7b). **Table 3** lists the advantages and drawbacks of each radar position.



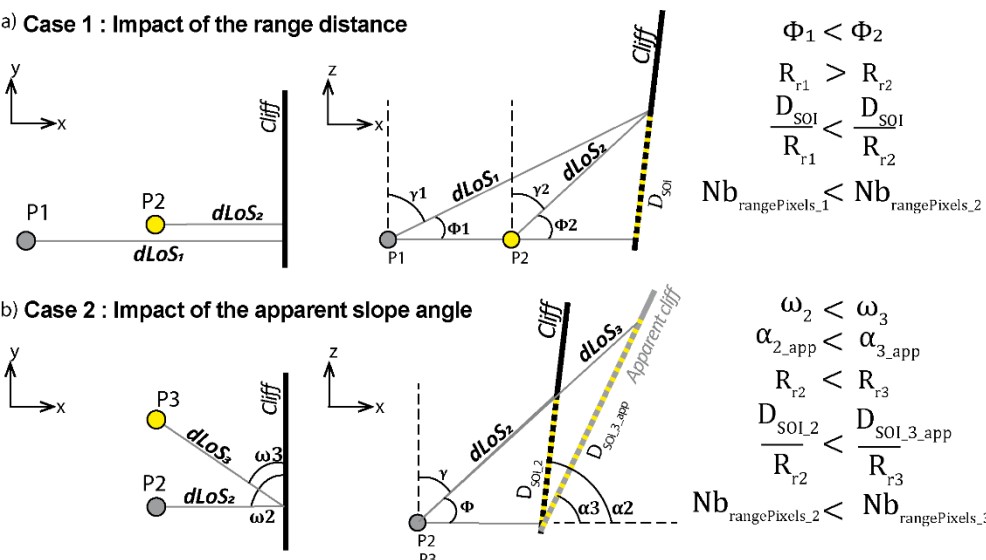

**Figure 7: Two scenarios of selection of the best GB-InSAR installation to get the less compression of the SoI in range. The best location is highlighted in yellow. a) Case 1: Installation near VS far from the monitored cliff. b) Case 2: Installation in front of VS from aside the monitored cliff. When looking aside, the apparent SoI length on slope in the radar direction will be longer so the information will be distributed in more pixels in range.**

**Table 3 : Advantages and drawbacks of each position presented in Figure 7.**

| POINT | Location | Advantages | Drawbacks | Resulting radar image |
|---|---|---|---|---|
| **P1** | • Range direction perpendicular to cliff strike<br>• Long range distance | Large illuminated area Dslope. | Poor range resolution, Compression of information in range. | |
| **P2** | • Range direction perpendicular to cliff strike<br>• Short range distance | Good range resolution, Important features can be distinguished. | Small illuminated area Dslope, Potentially more shadowing. | |
| **P3** | • Range direction not perpendicular to cliff strike (Decrease apparent slope)<br>• Short range distance | Smaller apparent dip and greater range. | Some important feature can be in shadow, LoS may not be parallel to the displacement, the recorded displacement value may be less than the real one. | |





### 2.3 Shadow, foreshortening and layover

Three geometrical notions must be reminded about radar geometry, because they can trigger noise and/or loss of information (Jensen, 2006):

- *Foreshortening*: Any terrain with a slope α inclined toward the radar (foreslope) result in a compression and a brightening of its surface in the radar image, also called foreshortening. Conversely, slopes inclined away from the radar (backslope) appear darker and elongated in the image (Figure 8a). The foreshortening factor $F_f$ can be defined as:

$$F_f = \sin(\alpha - \Phi) \quad (7)$$

- *Layover*: If the foreslope angle is greater than the look angle Φ, one can observe layover. The backscatter signal of the layover object will reach the radar receiver before the backscatter signal of the object located before. The information contained in the signal will be stored in the previous pixel in range of the image resulting in a layover distortion of the image which cannot be corrected. With a GB-InSAR, layover effects only occur in the case of an overhanging wall (Figure 8b).
- *Shadow*: An area hidden by a slope or by any other feature is not illuminated by the radar and will not be seen in the radar image, resulting in a loss of information. This can be for instance a deep valley or the ground behind a tall building (Figure 8c).

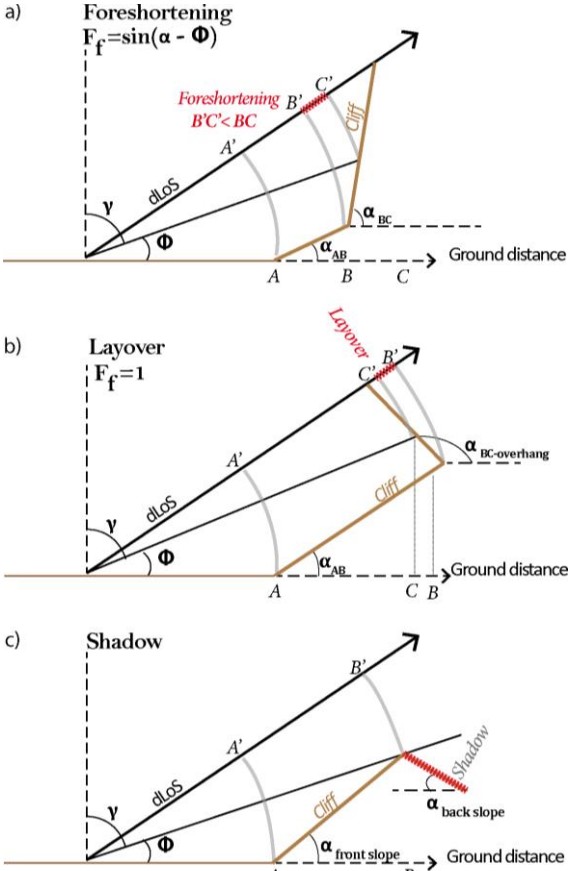

**Figure 8 : Illustration of geometrical artefacts in the case of GB-InSAR. a) Foreshortening, b) Layover, c) Shadow effects.**



## 3 Methodology

### 3.1 GB-InSAR constrains

Caduff et al. (2015) presents in a review a list of points to carefully consider when choosing the location for the GB-InSAR installation, including the visibility of the target area, the SoI, the foreshortening effects, the expected displacement (rate, direction, mechanism), the atmospheric influences and the technical constraints (hardware, setup, accessibility, …).

The MATLAB tool presented here is aimed at estimating some of those parameters and choosing the best location when several options are possible. The constrains estimated with the tool are the following:

- Range distance dLoS. The bandwidth BW must be adjusted based on this distance and in any case the target should not be further away by more than 4 or 5 km, depending on the GB-InSAR device, to avoid the risk of a too weak backscatter signal amplitude.
- Resolution in range and in azimuth. Once the potential location is found, it is worth estimating the resolution that will be obtained, as well as the extend of the surface illuminated by the radar.

- Areas in shadow and those subject to an important foreshortening.

### 3.2 Input parameters

#### 3.2.1 The DEM

The MATLAB tool requires as first input a Digital Elevation Model (DEM) in an ascii format, cropped around the area of interest. The MATLAB code converts it into a 3D point cloud, which is displayed in a window. For the use that will be made

of the DEM, a resolution of 5 m is sufficient. For each pixel of the grid, the dip and dip direction is calculated with the function "gradient" (MATLAB, 2023). Additionally, the user provides the coordinates of the middle of the area of interest and of the location where he considers installing the radar.

#### 3.2.2 Radar parameters

The parameters that will be estimated are related to some specificities of the radar itself and must be set up by the user before

starting the computation because they influence the resulting resolution as well as the extent of the illuminated area. Among those, there are the radar rail length L, the bandwidth BW and the beamwidth ε. Since the vertical beamwidth is limited in the case of the GB-InSAR, vertical and horizontal beamwidth are respectively selected by the user and denoted $ε_v$ and $ε_h$.

### 3.3 Estimation of parameters

#### 3.3.1 Foreslope and distance maps

Firstly, when installing the radar, the distance between the radar and the target surface must be estimated to not be greater than 4 or 5 km. Once the location of the radar $(x_{radar}, y_{radar}, z_{radar})$ is provided, a map is displayed giving for each point of coordinates $(x_{target}, y_{target}, z_{target})$ its distance to the radar defined by:





$$distance = \sqrt{\Delta x^2 + \Delta y^2 + \Delta z^2}, \quad (8)$$

with $\Delta x = x_{target} - x_{radar}$, $\Delta y = y_{target} - y_{radar}$, $\Delta z = z_{target} - z_{radar}$. Slopes facing the radar -or foreslope- can result
in foreshortening effect whereas slopes facing away -or backslope- will always be in shadow. That information is displayed in
a second map, where the foreslope points have a 1 value and the other points the value -1. To produce this map, the apparent
dip $\alpha_{app}$ is computed for each point (Addie, 1968):

$$\alpha_{app} = tan^{-1}(tan\,\alpha \,.\, sin(\omega_{LOS} - \omega_{slope})) \qquad (9)$$

With $\omega_{LOS}$ and $\omega_{slope}$ being respectively the orientation of the LoS toward north and the slope strike. From the apparent dip is
deduced the fore – and backslope as is:

$$\text{If } \alpha_{app} > 0 \rightarrow \text{Foreslope}$$

$$\text{If } \alpha_{app} < 0 \rightarrow \text{Backslope} \quad (10)$$

### 3.3.2 Radar footprint and illuminated area estimation

The radar footprint is an ellipse, within which the user selects a smaller area where to focus the radar acquisition. This
illuminated surface is selected by choosing the minimum and maximum azimuth $Az°_{min}$ and $Az°_{max}$ and the minimum and
maximum range $R_{min}$ and $R_{max}$ of the acquisition and should encompass the instable area to monitor.

Before installing the radar and starting the acquisition, it is worth checking the maximum possible radar footprint as well as
the illuminated surface according to the batch of azimuths and ranges selected by the user.

Once the radar and the target location are selected, the map is updated to display the footprint and the points illuminated by
the radar during the acquisition according to the parameters $R_{min}$, $R_{max}$, $Az°_{min}$ and $Az°_{max}$ selected by the user.

To do so, the coordinates of each point of the point cloud are converted from the global geographical coordinate system *global*
in a local coordinate system *local* whose frame origin is the center of the region of interest selected by the user. The X axis is
horizontal and perpendicular to the LoS direction and the horizontal Y axis is perpendicular to X. Z axis is a vertical unit
vector. Thus, the unit vectors of the *local* frame are:

$$\begin{pmatrix} \overrightarrow{X_{unit}} \\ \overrightarrow{Y_{unit}} \\ \overrightarrow{Z_{unit}} \end{pmatrix}_{local} = \begin{pmatrix} \dfrac{\Delta y_{global}}{\sqrt{\Delta y_{global}^2 + \Delta x_{global}^2}} & \dfrac{\Delta x_{global}}{\sqrt{\Delta y_{global}^2 + \Delta x_{global}^2}} & 0 \\ \dfrac{\Delta x_{global}}{\sqrt{\Delta y_{global}^2 + \Delta x_{global}^2}} & \dfrac{\Delta y_{global}}{\sqrt{\Delta y_{global}^2 + \Delta x_{global}^2}} & 0 \\ 0 & 0 & 1 \end{pmatrix}. (11)$$

Each point coordinates can be converted from the *global* geographical coordinate system to the new *local* coordinate

system by applying the translation of vector $\overrightarrow{dLoS}$ $\begin{pmatrix} \Delta x_{global} \\ \Delta y_{global} \\ \Delta z_{global} \end{pmatrix}$ followed by the rotation of matrix:

$$\Omega = \begin{pmatrix} a & -b & 0 \\ b & a & 0 \\ 0 & 0 & 1 \end{pmatrix}. \qquad (12)$$



The relation linking the coordinates of each point in the *global* geographical coordinate system and the new *local* coordinate system is:

$$\begin{pmatrix} x \\ y \\ z \end{pmatrix}_{local} = \Omega . \begin{pmatrix} x \\ y \\ z \end{pmatrix}_{global} + \overrightarrow{dLoS} \qquad (13)$$

$$\begin{pmatrix} x \\ y \\ z \end{pmatrix}_{local} = \underbrace{\begin{pmatrix} a & -b & 0 \\ b & a & 0 \\ 0 & 0 & 1 \end{pmatrix}}_{rotation\ matrix} . \begin{pmatrix} x \\ y \\ z \end{pmatrix}_{global} + \underbrace{\begin{pmatrix} \Delta x_{global} \\ \Delta y_{global} \\ \Delta z_{global} \end{pmatrix}}_{translation\ matrix} . \qquad (14)$$

The couple of values (a, b) is found by solving the system with the coordinates of the target point $\begin{pmatrix} 0 \\ 0 \\ 0 \end{pmatrix}_{local}$ in the local coordinate system:

$$\begin{pmatrix} 0 \\ 0 \\ 0 \end{pmatrix}_{local} = \begin{pmatrix} a & -b & 0 \\ b & a & 0 \\ 0 & 0 & 1 \end{pmatrix} . \begin{pmatrix} x_{target} \\ y_{target} \\ z_{target} \end{pmatrix}_{global} + \begin{pmatrix} \Delta x_{global} \\ \Delta y_{global} \\ \Delta z_{global} \end{pmatrix} \qquad (15)$$

All points of the point cloud are converted into the new local coordinate system with Equation 14. It is then possible to filter points that belong to the radar footprint, knowing $\varepsilon_v$ and $\varepsilon_h$, (Figure 9a). A point $P(x_{local}, y_{local}, z_{local})$ is within the footprint if:

$$|z_{local}| < \sqrt{H_\varepsilon^2 - \left(\frac{x_{local}}{L_\varepsilon}\right)^2}, \qquad (16)$$

with:

$$L_\varepsilon = dLoS(O) * \tan\varepsilon_h \qquad (17)$$
$$H_\varepsilon = dLoS(O) * \tan\varepsilon_v. \qquad (18)$$

Points belonging to the illuminated area are extracted in a second step. A point $P(x_{local}, y_{local}, z_{local})$ is within the illuminated area if (Figure 9b):

$$\begin{cases} P \in footprint \\ dLoS(P) \in [R_{min}; R_{max}] \\ z_{local} \in [W_{min}(P); W_{max}(P)] \end{cases} , \qquad (19)$$

with:

$$\begin{cases} W_{illu_{min}}(P) = dLoS(P) * \tan Az°_{min} \\ W_{illu_{max}}(P) = dLoS(P) * \tan Az°_{max}. \end{cases} \qquad (20)$$

Once the points illuminated by the radar are known, the mean normal plane vector $\vec{N}$ of this illuminated surface is estimated with a mean square method (Wolberg, 2006) and converted to mean slope dip $\alpha_{MEAN}$ and mean slop dip direction $\omega_{MEAN}$.



285

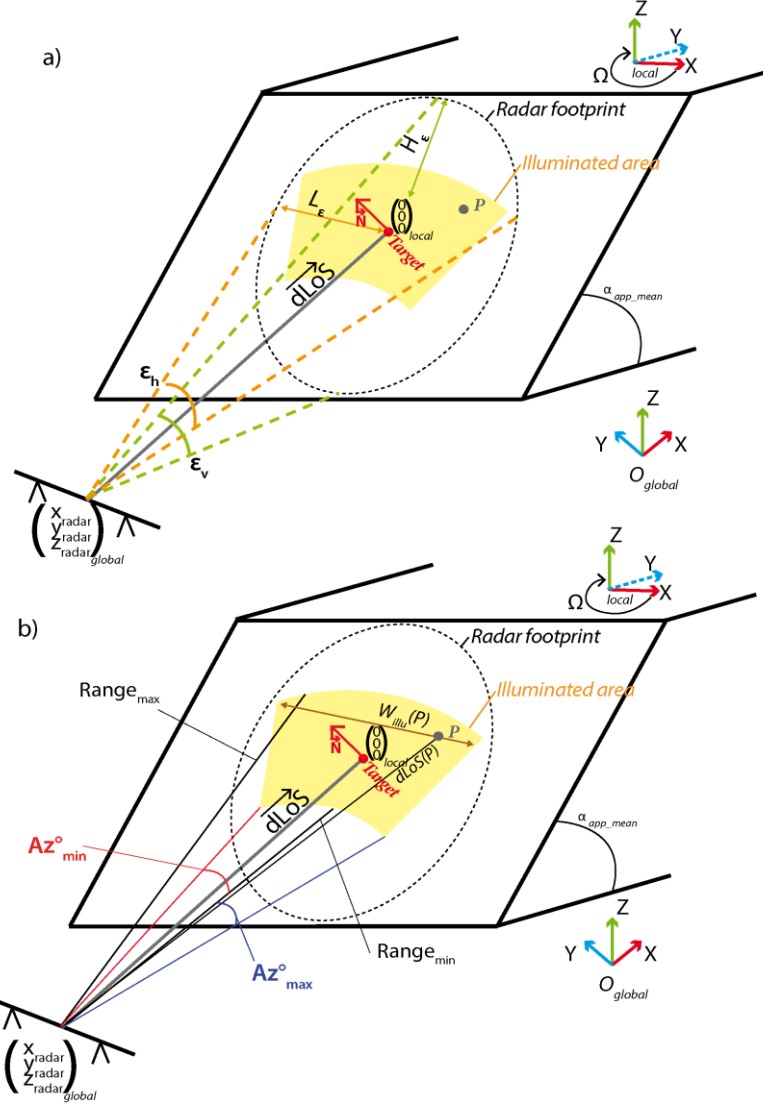

**Figure 9: Illustration of the change of coordinate frame from a global one $(X, Y, Z)_{global}$ to a local one $(X, Y, Z)_{local}$, centered on the target coordinates, aiming at highlighting the points of the point cloud within the radar footprint and the illuminated area. $X_{local}$ axis is horizontal and perpendicular to $\overrightarrow{dLoS}$, $Y_{local}$ axis is horizontal and perpendicular to $X_{local}$. a) Point $P$ is within the radar footprint because its coordinates answer the conditions defined in Equation 16. b) Point $P$ is illuminated because its coordinates answer the conditions defined in Equation 19.**

290

### 3.3.3    Range and azimuthal resolution map

Equations 2 and 5 are applied on each point of the point cloud to estimate the range and azimuthal resolutions and the result is displayed in two distinct maps.



### 3.3.4    Foreshortenings and layover map

The mean plan dip direction $\omega_{MEAN}$ and dip $\alpha_{MEAN}$ being known, the latter is converted in apparent dip from the radar position $\alpha_{MEAN\_app}$, and the foreshortening for this mean plan $F_{f-MEAN}$ is calculated by applying Equation 7, giving a value comprised between [0;1], 0 for no foreshortening and 1 for the beginning of layover. In addition, the foreshortening degree is calculated for each point of the point cloud.

It is then possible to estimate for each point if it will be affected by a stronger or a weaker foreshortening than the mean slope plan by subtracting the mean plan foreshortening $F_{f-MEAN}$ from the foreshortening calculated at each point. Such a map is also one of the outputs of the tool. A negative or positive value implies respectively a weaker or stronger foreshortening at that point than the mean one.

## 4    Results

### 4.1    Tool interface

The application coded in MATLAB is presented in a graphical user interface (Figure 10**Erreur ! Source du renvoi introuvable.**). The user first selects the DEM to import and convert into a 3D point cloud, and gives the coordinates of the location where he considers installing the radar and the coordinates of the middle of the of the region of interest. It can be found directly on the point cloud or from a GPS measurement performed on the field. Some radar parameters, such as the horizontal and vertical beamwidths, the bandwidth and the rail length used for focusing the signal must also be given before starting the computation (Table 4). For filtering pixels within the area illuminated during the processing, the user can also enter the minimum and maximum ranges and azimuths.

At the end of the processing, the different maps are displayed in 3D and can be stored in a text file to be reused later.

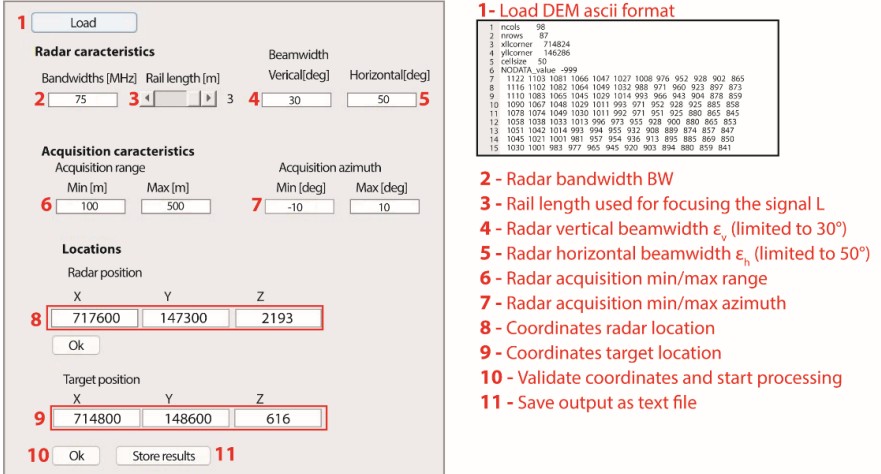

**Figure 10: Interface of the MATLAB tool and the format of the DEM grid to import, in a text file format.**





**Table 4 : Characteristics of the LiSA GB-InSAR used in this study.**

| Radar system | Rail length L [m] | Antenna type | Maximal horizontal/vertical beamwidth $\varepsilon_h/\varepsilon_v$ [°] | Radar band | Central pulse frequency f [GHz] | Bandwidth BW [MHz] | Measurable phenomena max speed [mm/h] |
|---|---|---|---|---|---|---|---|
| **Lisalab** | 3 | Horn antenna | 30 / 50 | Ku-band | 17,2 | [50-175] | 176 |

## 4.2   Case study location

To validate the results obtained with this tool, the later has been tested on (1) a simple synthetic cliff created in Cloud Compare,
facing North and with a progressive slope and a backslope (Figure 11, Figure 1) and (2) two real cliffs where GB-InSAR
monitoring campaigns were conducted (Table 5, Figures 12 and 13).

**Table 5 : The three study cases with their DEM characteristics and the radar parameters chosen.**

| Case study | Geographical coordinate system | Radar location | Target location | dLoS [m] | BW [Mhz] | Min/Max Ranges [m] | Min/Max Azimuths [°] |
|---|---|---|---|---|---|---|---|
| **1- Synthetic dataset** | - | x – 0 y – 100 z – 1 | x – 0 y – -88 z – 20 | 20 | 175 | 180/205 | -12/12 |
| **2- Cima del Simano** | CH1903 LV03 | x – 714800 y – 148600 z – 616 | x – 717600 y – 147300 z – 2193 | 3500 | 75 | 3100/4200 | -9/9 |
| **3- La Cornalle** | CH1903 LV03 | x – 547000 y – 149000 z – 230 | x – 548000 y – 150000 z – 685 | 300 | 175 | 140/250 | -4/4 |



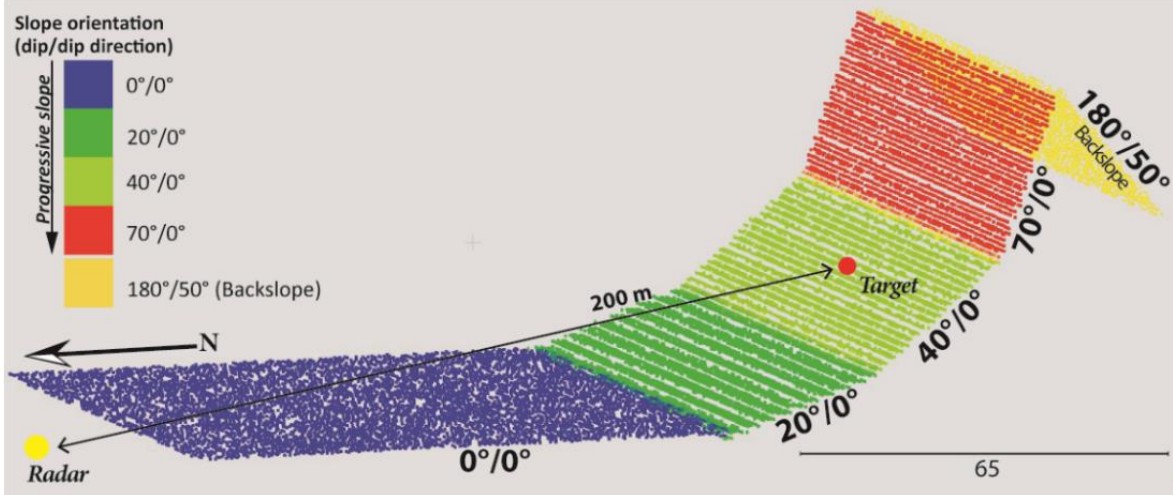


**Figure 11 : Synthetic dataset: a cliff facing North with a progressive slope dip and a backslope.**

### 4.2.1    Case study 1: Cima del Simano instability monitoring

Cima del Simano is a deep-seated landslide located in the Ticino canton in Switzerland (Figure 12a). Satellite interferometry measurements highlight the presence of slow sub-vertical gravitational movement on top of the mountain, which motivated

the launch of a GB-InSAR monitoring campaign in 2021 (Wolff et al., 2022). A Lisalab GB-InSAR with a 3 m-rail was installed in the valley near a building belonging to the Acquarossa commune to have access to electrical power (Figure 12c). The acquisition is challenging because the radar is located at its range limit and the top of the cliff, at an altitude of 2500 m, witnesses strong atmospheric effects (Figure 12b).






**Figure 12 : First real studied site: Cima del Simano. a) Location. b) Radar and target location on Google Earth image. c) Lisalab GB-InSAR installation.**

### 4.2.2    Real case 2: La Cornalle cliff monitoring

La Cornalle cliff located in the Lavaux vineyard (East Lausanne, Switzerland, Figure 13a) is monitored with yearly LiDAR acquisitions since 2013 (Carrea et al., 2014, 2015). This steep cliff is affected by important erosion process and frequent rockfall events occur every year (Figure 13b). This site is monitored to estimate the role of the rock surface temperature and of the atmospheric erosion (Fei et al., 2023). In this topic, a GB-InSAR has been installed in August 2022 at the bottom of the cliff. The Lisalab rail has been set up on a flat wall near a vineyard factory to have access to electricity and very near the cliff

to get the best possible resolution (Figure 13c).



**Figure 13 : Second real studied site: La Cornalle cliff. a) Location. b) Radar and target location on Google Earth image. c) Lisalab GB-InSAR installation.**

## 4.3    Results for the three study sites

The obtained results are summarized in Table 6 for the three studied cases and some output maps are displayed in Figure 14 in the case of the synthetic dataset, Figure 15 in the case of Cima del Simano and in Figure 16 for La Cornalle cliff. The extra output maps are available in the appendices A to C.





**Table 6: Results obtained with the MATLAB tool for the three study sites.**

| Case study | Location | Range resolution [m] | Azimuthal resolution [m] | dLoS [m] | Mean dip | Mean dip direction | Mean foreshortening | Output maps |
|---|---|---|---|---|---|---|---|---|
| **0** | Synthetic dataset | 0.95 | 0.57 | 200 | 48° | 0° | 0.67 | Figure 14 |
| **1** | Cima del Simano | 2.0 | 10.4 | 3351 | 51° | 344° | 0.38 | Figure 15 |
| **2** | La Cornalle | 0.92 | 0.64 | 267 | 61° | 201° | 0.81 | Figure 16 |

### 4.3.1 Synthetic dataset (Figure 14, Appendix A)

With a LoS distance of 200 m, the range and azimuthal resolutions are respectively 0.95 m and 0.57 m at the target location.
The calculated apparent mean dip is 48° because the illuminated area encompasses a major part of the slope with a dip of 40° and a little part of the slope with a dip of 70°. The mean foreshortening of the SoI is 0.67 (Figure 14e) but the points located on the slope with a dip of 40° are less affected by foreshortening than those located on the slope with a dip of 70° (Figure 14f).

### 4.3.2 Cima del Simano (Figure 15, Appendix B)

The LoS distance is almost at the limit of what is acceptable to have a backscattered signal (3351 m at the target location and
3900 m near the crest). Such a distance decreases considerably the azimuthal resolution, which varies between 8 m and 13 m along the range of the illuminated slope (Figure 14c) for a range resolution of 2.0 m (Figure 14d). The SoI is affected by a constant and low foreshortening of 0.38 (Figure 14e, f).

### 4.3.3 La Cornalle (Figure 16, Appendix C)

The LoS distance is 267 m resulting in a good resolution in azimuth varying between 0.3 m and 0.7 m along the SoI (Figure
16c) and in range between 0.8 m and 1 m (Figure 16d). The slope is very steep, the apparent slope dip is 73°. Consequently, the radar image is affected by an important foreshortening of 0.81 (Figure 16e). In the middle of the slope illuminated by the radar (Figure 16a), one can see a little terrace in backslope and affected by shadowing (Figure16b). The slope below this terrace is less steep than the illuminated area mean plan oriented 61°/267° while the one located above is steeper. The lower part is less affected by foreshortening than the upper part (Figure 16f).




**Figure 14: Output maps for the synthetic dataset. a) Illuminated area and footprint for the chosen parameters. b) Front VS back slope. The back slope corresponds to areas in shadow. c) The azimuthal resolution increases with the distance to the radar. d) The range resolution increases with the slope dip but in a same slope degree, it decreases from near- to far-range. e) Zoom in the illuminated area for which the mean plan is estimated. The mean dip of the area is 48° with a mean foreshortening of 0.67. f) Foreshortening degree compared to the mean plan one $F_{f\text{-MEAN}}$. Smoother slopes are less affected by the foreshortening than the mean plan. The steeper ones are more affected. The other output maps are presented in the Appendix A.**







**Figure 15: Output maps for the first dataset, Cima del Simano. a) Illuminated area and footprint for the chosen parameters. b) Front VS back slope. The back slope corresponds to areas in shadow. c), d) Azimuthal and range resolutions. Since the chosen bandwidth is 75 MHz to be sure to record the backscattered signal, the resolutions are poor. Only a monitoring of major volume instabilities is relevant here. e) Zoom in the illuminated area for which a mean plan fitting is estimated. The mean dip of the area is 51° with a mean foreshortening of 0.38. f) Foreshortening degree compared to the mean plan one. The lower part is more affected by foreshortening that the top of the mountain, which is the area of interest. The other output maps are presented in the Appendix B.**







**Figure 16: Output maps for the second dataset, La Cornalle cliff. a) Illuminated area and footprint for the chosen parameters. b) Front VS back slope. The back slope corresponds to areas in shadow. c), d) Azimuthal and range resolutions. Since the radar is only at a distance of 260 m, the selected bandwidth is 175 MHz to have a good range resolution. e) Zoom in the illuminated area for which a mean plan fitting is estimated. The mean dip of the area is 61° with a mean foreshortening of 0.81. f) Foreshortening degree compared to the mean plan one. The other output maps are presented in the Appendix C.**



## 5    Discussion

### 5.1    Comparison for three different radar locations

Figure 7 presented the impact of the radar location on the resulting image. To verify those concepts, three radar locations, P1, P2 and P3, have been selected on the study site of Cima del Simano to monitor the same region of interest. Their output
characteristics are summarized and compared in Table 7. Locations P1 and P2 are located in front of the cliff and along the direction parallel to the slope dip; P1 being further away from the target area than P2. P3 is located almost at the same distance of the cliff than P2 (P3 is 113 m further away from the target than P2) but looks at the target area from aside.

The results are coherent with what is expected from Table 3. An acquisition from P2 gives a better range resolution (2.01 m) compared to P1 (2.14 m). Those two points being located in front of the cliff, their apparent slope dip corresponds to the real
one (64°), while the apparent slope dip for P3 is lower (60°) reducing slightly the range resolution (2.00 m). One could conclude that the best location for the acquisition is P3. Nonetheless this position triggers here more areas in shadow (backslope) leading to a loss of information. Furthermore, the gravitational movements are often expected to follow the slope dip direction (Pedrazzini et al., 2010; Dehls et al., 2010). Looking to the target area from aside as with P3, the LoS direction is not parallel to the slope, the registered displacement may be lesser than the real one (Colesanti and Wasowski, 2006; Dai et al., 2022).
The illuminated area is wider in the case of P1 than with P2 or P3. If the unstable area to monitor is very large, P1 can be advantageous. Since the LoS remains smaller than 4 km (3.9 km); the backscattered signal is still registered from P1.

Thus, the best installation location highly depends on the purpose of the acquisition and the area of interest.

- If the main goal is to monitor a large area in order to detect unstable zones and estimate an average displacement rate such as in Carlà et al. (2019), a position far from the monitored cliff (similar to P1) should be considered. The tool
can help checking that the LoS distance remains smaller than 4 or 5 km and estimating the extent of the illuminated area. But one must be aware that the resulting range and azimuthal resolutions of the radar image increase.

- If the main goal is to get the best resolution at the expense of the illuminated area size, one should try locating the radar closer to the monitored cliff (similar to P2 and P3). It is the case for example when one tries to define the kinematic behavior (Frattini et al., 2018) or to assess the susceptibility to fail (Jaboyedoff et al., 2012) of massive
rock instabilities. The tool helps checking which location (close to the cliff, in front of or aside the cliff) gives the best resolution while avoiding having the SoI in the shadow.




**Table 7 : Comparison of three different radar locations and the impact on their corresponding radar image. Advantages and drawbacks for each position are highlighted.**

| Corresponding position in Figure 7 Test case | P1 In front of the cliff Far from the cliff | P2 In front of the cliff Close to the cliff | P3 Aside the cliff |
|---|---|---|---|
| Coordinate system | | CH1903 - LV03 | |
| Target location | | [717600 ; 147300 ; 2193] | |
| Radar location | [714108 ; 148128 ; 750] | [715400 ; 148700 ; 635] | [715201 ; 147193 ; 556] |
| Backslope |  |  |  |
| Illuminated area |  |  |  |
| dLoS | 3927 | 3282 | 3150 |
| app. Dip | 64° | 64° | 60° |
| Azimuthal resolution | 11.64 | 9.89 | 8.72 |
| Range resolution | 2.14 | 2.01 | 2.00 |
| Advantages | - Wider extend of the illuminated surface | -Better resolutions than P1 | -Better resolutions than P1 and P2 |
| Drawbacks | -Compression of information in range -dLoS limit for recording backscattered signal | -Smaller illuminated surface than P1 - More shadow than P1 | -More shadow than P1 and P2 - Measured displacement along the LoS may be lower than the real one |


## 5.2   Tool limitations

Since the input of the program is a DEM converted into a 3D grid, the overhanging slopes, which are those subject to layover cannot be detected.  To overcome this limitation, a suggestion could be to use a point cloud acquired from the ground with a





LiDAR (Abellán et al., 2014) or by photogrammetry (Eltner and Sofia, 2020). But this comes with other problems, such as

the potential occlusions (Sturzenegger et al., 2007) and noises due to the presence of vegetation and which can bias the calculated dip and dip direction of the slope.

## 6   Conclusion and further development

This paper presents a new MATLAB tool dedicated to the estimation of the main characteristics of a Linear GB-InSAR acquisition in terms of resolutions, illuminated area and potential geometrical distortions. It is based on the transposition of

satellite radar geometry – which is well-documented- to the ground radar geometry. This tool can ease the process of finding the best radar installation location, which will provide the best monitoring results, when several are considered. This best location depends on the purpose of the acquisition campaign. Finding the best location is indeed not always easy since radar imagery is drastically different from optical geometry we are generally used to.

If the purpose is to monitor a large area and to delimitate the moving zone, the radar should be installed far from the cliff,

using the MATLAB tool to check that the LoS distance remains below 4 or 5 km, depending on the GB-InSAR device. Contrariwise, if the purpose is to characterize the displacement gradient, one will try optimizing the resolution while keeping the LoS as parallel as possible to the displacement vector and avoiding the shadowing areas.

Nevertheless, the radar acquisition characteristics are often not the only thing to consider when choosing the best location. Most of the time, the electricity access and an easy installation on a flat surface, as well as the expected instability movement

direction, reduce the choices (Caduff et al., 2015).

The tool could be improved and extended to the other GB-InSARs of type ArcSAR or rotary RAR (Pieraccini et Miccinesi 2019) and for the estimation of satellite InSAR images characteristics in order to select the best ascending or descending orbit acquisition before starting the downloading and treatment of the images which can also be a long and laborious work (Berardino et al. 2002; Mancini et al. 2021).

## 7   Appendices



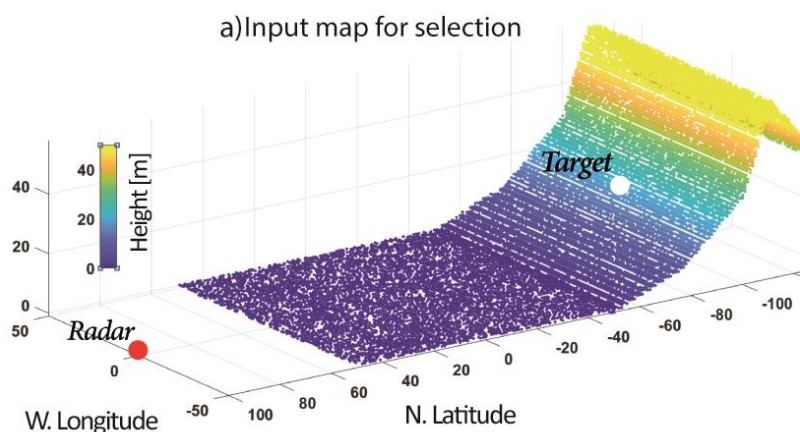

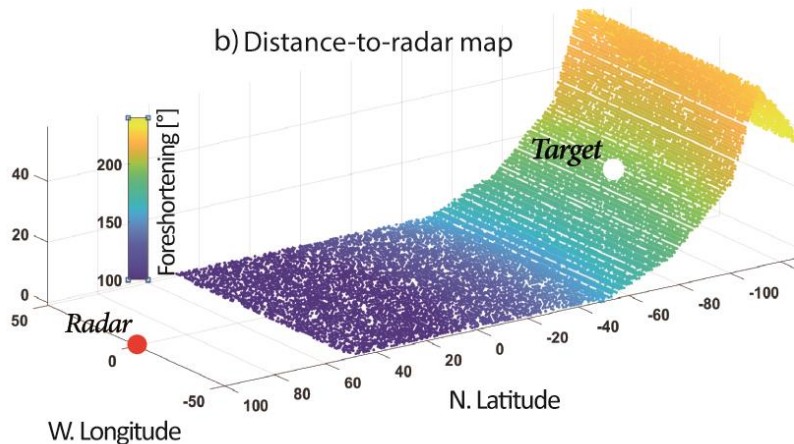

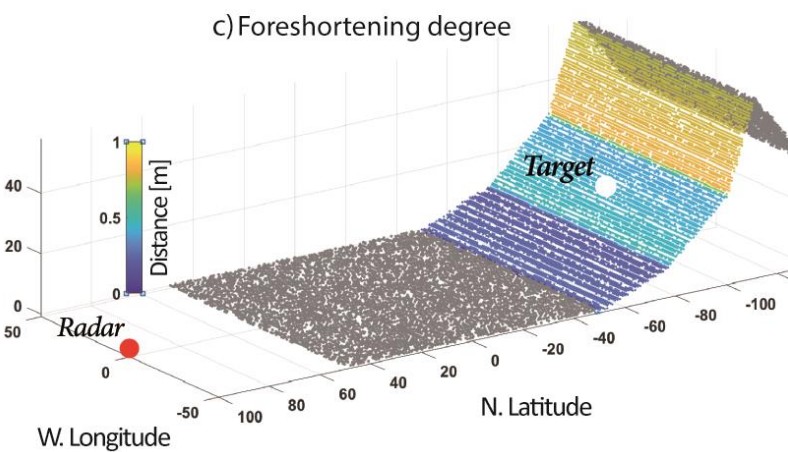

**Appendix A: Additional output map for the synthetic dataset. a) The input map with the altitude, aimed at choosing the radar and target location b) The distance-to radar map. The distance between the radar and the target should not be greater than 4 km. c) The**
**foreshortening map, after the Equation 7.**







**Appendix B: Additional output map for the first dataset, Cima del Simano. a) The input map with the altitude, aimed at choosing the radar and target location b) The distance-to radar map. The distance between the radar and the target should not be greater than 4 km. c) The foreshortening map, after the Equation 7.**





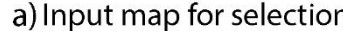

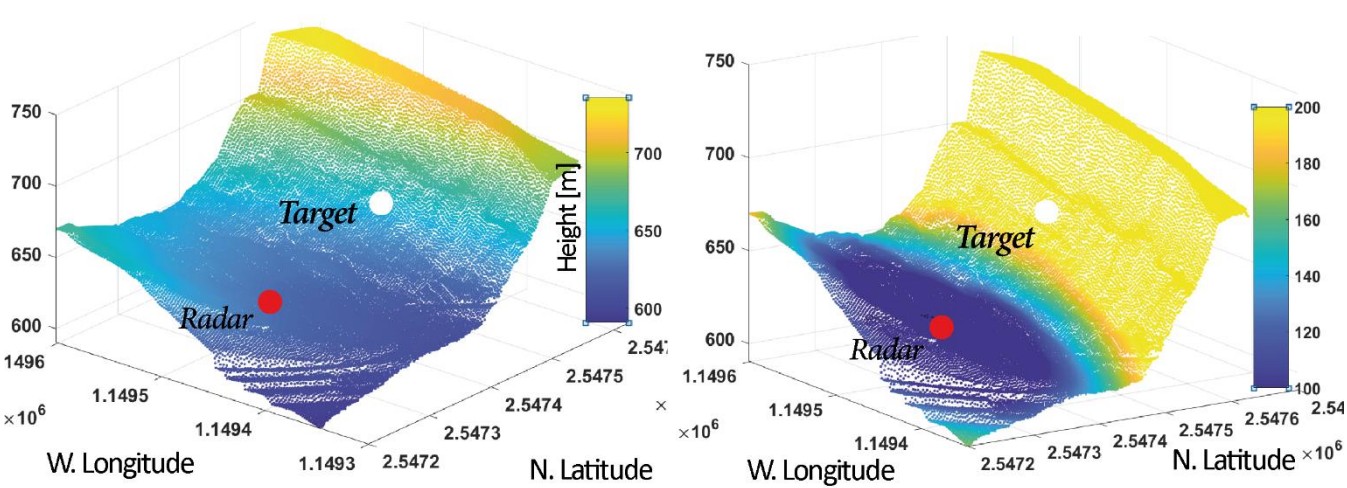

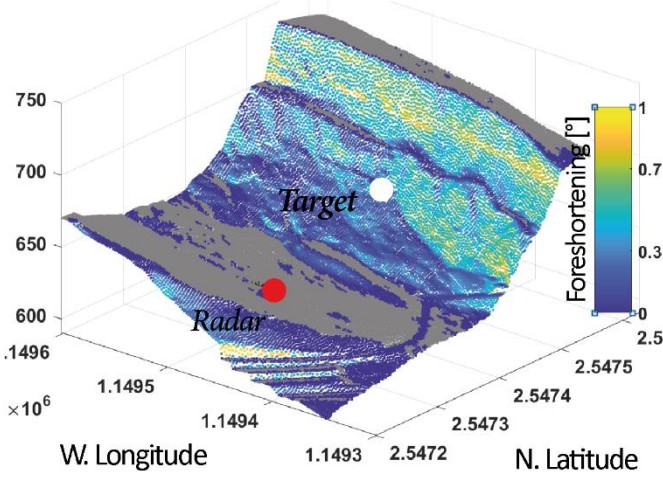

**Appendix C: Additional output map for the second dataset, La Cornalle. a) The input map with the altitude, aimed at choosing the radar and target location b) The distance-to radar map. The distance between the radar and the target should not be greater than 4 km. c) The foreshortening map, after the Equation 7.**

## 8    Code and data availability

All raw data can be provided by the corresponding authors upon request. The code and the input point clouds are available on Github. (https://github.com/charlottewolff/GB-PAR)



## 9    Author contributions

MHD had the tool idea; CW wrote and tested the code; CW created the synthetic dataset and performed the acquisition of the real datasets; CR provided documentation for comparing satellite and GB- techniques; CW wrote the manuscript draft and
edited the manuscript; MHD, CR, MJ reviewed the manuscript.

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
