# Peer review of "A new tool for the estimation of Ground-Based InSAR acquisition characteristics before starting installation and monitoring survey"

_EGUsphere, 2023_

## Referee Comment (RC1)

**Review of egusphere-2023-2489**

In the manuscript "A new tool for the estimation of Ground-Based InSAR acquisition characteristics before starting installation and monitoring survey", Wolff et al. present a software tool to estimate different acquisition characteristics of GB-SAR installation sites based on a Digital Elevation Model of the study area. The motivation behind is to provide a tool to find the best installation location of the GB-SAR, in the case that several potential sites are available. The best installation site depends on the geometrical resolution, image distortions and backscatter intensity. They test the tool with a synthetic and two real study cases.

While I think that such a tool can be quite helpful to GB-SAR users, I think that the paper should not be published without major revisions. Below, I provide general and specific comments and suggestions for improving the manuscript that should be addressed prior to publication.

*Major concerns:*
- Various descriptions / visualizations / statements in the manuscript are incorrect:
  - In Figure 1, the authors visualize the emitted signal of satellite SAR systems as waveforms with uniform frequency. This is incorrect. Satellite SAR systems emit frequency modulated signals in order to achieve a fine slant range resolution after pulse compression.

  - P5, L94: "The duration of one sequence of frequency variation is also called pulse length τ and is linked to the BW by the following equation ..". First, sweep duration or sweep length instead of pulse length is commonly used to describe the length of a frequency sweep of FMCW radars. Furthermore, the equation does not hold true! The inverse of the sweep duration equals the smallest measurable frequency shift for FMCW radars, and the time resolution corresponding to this frequency resolution is the inverse of the bandwidth. The latter is implicitly made use of in Equation 5.
  I guess you mixed this up with pulsed frequency modulated radars, for which the effective pulse length (after pulse compression) is the inverse of the chirp bandwidth.

  - P5, L101: "The range resolution is inversely proportional to the real antenna length L_real". No, not the range resolution but the azimuth resolution is inversely proportional to the antenna length!

  - Figure 2: The incidence angle for the satellite case is not the sum but the difference of look and slope angles.

  - Figure 3: As I understand it, the yellow cells are supposed to visualize the distortion within the specific acquisitions. Furthermore, I am interpreting shrinked cell sizes in Y-direction as a result from foreshortening in the radar images. If this is correct, why is the cell size in Y-direction decreasing along the Y-axis in (d)? Actually the foreshortening would decrease with increasing look angle along the slope. Also i don't understand what the following in the Figure caption means: "distance between two horizontal lines increased along Y due to an increase of the range resolution" Actually the range resolution is getting better (means smaller resolution) along Y, isn't it? Lastly, It seems that the satellite SAR image is distorted in X-direction, showing as increased X cell sizes in the center of the image. Why is that?
  Maybe I'm misinterpreting the figure, however in that case the figure should maybe be reconsidered.

  - P8, L136: "The satellite InSAR image footprint is thus rectangular". How is the shape of the satellite SAR footprint related to the azimuth resolution? Also on the contrary, you write on P4, L78 "The radar footprint on the ground is an ellipsoid"

  - Figure 7: how can omega_2 be smaller than omega_3? Also alpha_3 is larger than alpha_2.

- Conclusion: i think the usage of the proposed tool should be motivated and promoted better in the conclusion.. including that the tool is able to identify shadow areas and foreshortening etc for different radar positions, which is helpful to know beforehand.

  The advice given here regarding placement of the GB-SAR for large area monitoring is not a new finding of the paper, and determining the distance of the GB-SAR to the area of interest is not a difficult task, you don't necessarily need your tool for that. I think the focus here should be much more on the real advantages of using your proposed tool.

- P9, L158: "A good balance between an acceptable resolution and backscattered enough signal must be found (Figure 6a, b)".You describe that the maximum distance at which the GB-SAR can be deployed is 4-5 km, and that radar parameters have to be adjusted with regard to the distance. How do you choose these parameters in practice? Are you testing different parameters and comparing the results? Or are there specifications in the GB-SAR manual? Have you thought about including in your tool a rough estimate of the backscatter intensity of the point of interest with regard to the radar

position, in order to find the appropriate radar parameters at a certain distance. Then the range resolution can be automatically updated with this information.

- The structure of the manuscript and also the formatting in certain cases make it difficult to read. Some examples:
  - P17, L307: The following paragraph is completely redundant, as the necessary input parameters and handling of the tool has already been described in Section 3.2.
  - P16, L292: Section 3.3.3 consists of only one sentence, which is not good practice.
  - P8, L137: Why is this sentence in bold letters and centered?

- I find the manuscript to suffer from poor usage of the English language. Proofreading by a native speaker would improve the reading of the manuscript considerably, I believe. Some examples:
  - P2, L48: "...consisting in a radar measuring head translating along a rail."
  - P4, L84: "...the amplitude of the signal sent must be important to reach the Earth surface and to be backscattered with enough intensity to be recorded by the radar receiver"
  - P9, L159: ".. backscattered enough signal.."
  - P10, L174: "A good monitoring is when the information related to this SoI is distributed in the maximum of pixels in range and not compressed only in a few ones"
  - P13, L226: "… middle of the area …"
  - P22, L371: " … the radar image is affected by an important foreshortening …"

*Minor comments:*
- P1, L24: "..it was dedicated". It is still, inter alia, dedicated to studying small movements phenomena. So usage of "has been dedicated" would fit better in my opinion.
- P4, L62: Section 2.1.1 I think it's confusing why range and azimuth directions are not introduced for GB-SAR here.
- P4, L69: "Those angles" In my opinion, it's not good practice to refer to the paragraph title like that.
- P5, L92: "signal emitted is of lower intensity". Compared to what?
- P5, L94: Figure 1e and 1g should be 1f and 1h, respectively.
- P10, L182: The table and its contents could be explained in much more detail, e.g. why the detected displacement is lower than the real displacement.. this can be, by the way, the case in every radar acquisition geometry.
- Table 3: Different font sizes used here
- P14, Equation (9): Formatting..
- P14, L251: " … should encompass the instable area to monitor."  Shouldn't be some stable area also be included as reference?
- P15, Eq. 19, dLos in [Rmin,Rmax] .. does this definitively hold true in case of local topography?  Or is the "mean plane" used here?
- P17, L296: What's the "mean plane"? Hasn't been introduced before.
- Table 5: dLos of synthetic test should be 200m, I guess. Furthermore, how can dlos in case 3 be outside of the range limits?
- Table 6: i did not get why is there a difference between the dLoS in Table 5 and Table 6?
- P26, L411: LoS is usually referred to as a vector or direction, not distance.
- Figure A: Colorbar labels of foreshortening and distance maps are mixed up
- Figures A/B/C: As I understand it, the foreshortening degree should be without unit? Instead of given in degrees..
- P35, L572: gradientm?
- P36, L623: Where was this published?

---

## Author Comment (AC1)

Dear reviewer,

Thank you sincerely for dedicating your time to thoroughly review our article and for providing such an insightful and constructive analysis. Your detailed feedback, highlighting both major issues and missing details, was much valuable. We carefully considered each of your comments and implemented the suggested corrections. Please find our responses to your comments below. Once again, we truly appreciate your valuable input.

I-    Major concerns:

• Various descriptions / visualizations / statements in the manuscript are incorrect:

◦ In Figure 1, the authors visualize the emitted signal of satellite SAR systems as waveforms with uniform frequency. This is incorrect. Satellite SAR systems emit frequency modulated signals in order to achieve a fine slant range resolution after pulse compression.

Indeed, we wanted to keep the explanations of InSAR range resolution as simple as possible, omitting this point. Figure 1 was corrected:

- The satellite signal is now also frequency modulated
- The nomenclature is improved, distinguishing BW for satellite and GB-InSAR (BW$_{FMCW}$ and BW$_{FMPR}$)

◦ P5, L94: "The duration of one sequence of frequency variation is also called pulse length τ and is linked to the BW by the following equation ..". First, sweep duration or sweep length instead of pulse length is commonly used to describe the length of a frequency sweep of FMCW radars.

In table 1, lines are added for explaining terms FMCW and FMPR (frequency modulated pulsed radar). $\tau$   is distinguished between $\tau_{FMPR}$ (Pulse length) and $\tau_{FMCW}$ (sweep length).

◦ Furthermore, the equation does not hold true! The inverse of the sweep duration equals the smallest measurable frequency shift for FMCW radars, and the time resolution corresponding to this frequency resolution is the inverse of the bandwidth. The latter is implicitly made use of in Equation 5. I guess you mixed this up with pulsed frequency modulated radars, for which the effective pulse length (after pulse compression) is the inverse of the chirp bandwidth. →For clarity and simplicity, all the information is gathered with the corresponding references: *It is ground-geometry dependent, linked to the incidence angle θ, the speed of light c and the pulse length $\tau_{FMPR}$ or sweep length $\tau_{FMCW}$ according to the following relations (Henderson and Lewis, 1998; Jensen, 2006; Mahafza, 2000; McCandless and Jackson, 2004):*

$$R_{r,satellite} \quad = \frac{\tau_{FMPR}\,c}{2\sin\theta} = \frac{c}{2\,BW\,\sin\theta} = \frac{c}{2\,BW\,\sin(\Phi - \alpha_{app})} \qquad (3)$$

$$R_{r,GB-InSAR} = \frac{\tau_{FMCW}\,c}{2\sin\theta} = \frac{c}{2\,BW\,\sin\theta} = \frac{c}{2\,BW\,\sin(90 - \alpha_{app} + \Phi)} = \frac{c}{2\,BW\,\cos(\Phi - \alpha_{app})} \qquad (4)$$

◦ P5, L101: "The range resolution is inversely proportional to the real antenna length L_real". No, not the range resolution but the azimuth resolution is inversely proportional to the antenna length!

This was an error. We corrected the range to azimuthal.

◦ Figure 2: The incidence angle for the satellite case is not the sum but the difference of look and slope angles. This error was changed in the figure, and we checked that it is correct in the corresponding equation.

◦ Figure 3: As I understand it, the yellow cells are supposed to visualize the distortion within the specific acquisitions. Furthermore, I am interpreting shrinked cell sizes in Y-direction as a result from foreshortening in the radar images. If this is correct, why is the cell size in Y-direction decreasing along the Y-axis in (d)? Actually, the foreshortening would decrease with increasing look angle along the slope. Also i don't understand what the following in the Figure caption means: "distance between two horizontal lines increased along Y due to an increase of the range resolution" Actually the range resolution is getting better (means smaller resolution) along Y, isn't it? Lastly, It seems that the satellite SAR image is distorted in X-direction, showing as increased X cell sizes in the center of the image. Why is that? Maybe I'm misinterpreting the figure, however in that case the figure should maybe be reconsidered. The foreshortening is the slope length of the surface in the image: It is shortened for radar compared to the optical image. This is added in the figure for clarity.

For Radar, along Y, the range resolution decreases (because the look angle increases). The resolution is thus better in the far range. Two consecutive lines are thus comprised in the same pixel in the near-range (and are not resolved) but are in two different pixels in the far-range (they are resolved). This is figured out in the image by an increase of the distance between two consecutive lines along the Y axis (figure c and d). We have added that one square in the image corresponds to one pixel.

Furthermore, we realized that it may not be clear for the reader the satellite radar image for the cliff since a satellite acquire from the sky and not from the ground as suggested in figure 3a). We decided to divide the figure in 2 parts: (1) Acquisition with camera or GB-InSAR from the ground of a cliff and (2) Acquisition of a flat surface with a satellite InSAR from the satellite orbit.

The figure title is changed for clarity and accounting of those changes:

*Figure 1: Comparison of the view of a surface. a) Real view in a parallel projection with the camera or radar acquiring the image from the ground, b) Optical image taken with a camera from the ground. The slope is not compressed. The resolution R increases with Z. Consequently, the distance between two consecutive horizontal lines decreases along Z, c) GB-InSAR SAR image (after Tapete et al., 2013). Raz increases and Rr decreases along Y. Consequently, the distance between two consecutive horizontal lines increases along Z. d) Real view in a parallel projection with the satellite radar acquiring the image from the satellite orbit. e) Satellite SAR image. Raz is constant along Y while Rr decreases. Consequently, the distance between two consecutive horizontal lines increases along Y. In the case of the radar images (c, e), the slope is compressed compared to the optical image, due to the foreshortening effect.*

◦ P8, L136: "The satellite InSAR image footprint is thus rectangular". How is the shape of the satellite SAR footprint related to the azimuth resolution? Also on the contrary, you write on P4, L78 "The radar footprint on the ground is an ellipsoid". This was indeed not clear. It is not the footprint but the representation of the slope in the radar image. This sentence was removed since it is implicitly presented into the figure2.

◦ Figure 7: how can omega_2 be smaller than omega_3? Also alpha_3 is larger than alpha_2. → The explanation is improved :

The text is completed to explain:

*The apparent slope angle can be reduced by placing the radar aside instead of in front of the measured slope and by applying the Equations (Addie, 1968):*

$$\alpha_{app} = \tan^{-1}(\tan\alpha \times \sin\omega)\ (7)$$

$$\omega = \omega_{LOS} - \omega_{slope},\ \ (8)$$

*with ω the angle between the slope direction and the LoS direction, and ω $_{LOS}$ and ω $_{slope}$ being the orientation of the LoS toward north and the slope strike, respectively.* **Erreur ! Source du renvoi introuvable.** *lists the advantages and drawbacks of each radar position.*

The image is updated/ corrected:                                                    $\omega_2 > \omega_3$.

The legend is updated: Figure 7:

*Two scenarios of selection of the best GB-InSAR installation to get the less compression of the SoI in range. The best location is highlighted in yellow. a) Case 1: Installation near VS far from the monitored cliff. b) Case 2: Installation in front of VS from aside the monitored cliff. When looking aside,* $\boldsymbol{\alpha_{app}}$
*is smaller according to Equation 7 and the apparent SoI length on slope in the radar direction is longer so the information is distributed in more pixels in range.*

• Conclusion: i think the usage of the proposed tool should be motivated and promoted better in the conclusion.. including that the tool is able to identify shadow areas and foreshortening etc for different radar positions, which is helpful to know beforehand. The advice given here regarding placement of the GB-SAR for large area monitoring is not a new finding of the paper, and determining the distance of the GB-SAR to the area of interest is not a difficult task, you don't necessarily need your tool for that. I think the focus here should be much more on the real advantages of using your proposed tool. We think that the interest of the paper is dual:

(1) Describing the major differences between GB-InSAR and satellite InSAR because when I started developing the tool we met difficulties finding papers describing GB-InSAR acquisition characteristics.
(2) Presenting the novel MATLAB tool which gather in the same place a set of valuable maps for evaluating what results we can obtain before starting the monitoring campaign and checking that the distance is to target is ok AND not into shadow or affected by a strong foreshortening. This tool was developed to answer our needs in natural hazard monitoring and we thought it could be useful also for others to have a tool doing all the calculations for them before the radar installation.

The conclusion was modified to support these two points:

This paper described the main features of a Linear GB-InSAR acquisition, emphasizing and comparing the significant differences from satellite radar acquisitions. While these distinctions are rarely addressed in the literature, they are crucial considerations for anyone initiating a GB-InSAR monitoring campaign.

The paper introduces in a second step a novel MATLAB tool designed for the estimation of the characteristics of Linear GB-InSAR acquisitions. This tool generates a set of valuable maps, including the radar-to-target distance, range and azimuthal resolution, foreshortening degree, and shadowing maps in a single operation. The main purpose is to streamline the search for the optimal radar installation site, which guarantees the most effective monitoring results when multiple options are considered.

Since the determination of the ideal location varies depending on the objectives of the acquisition campaign, providing comprehensive information critical for selection simplifies the sensitive choice for the most suitable site.

If the purpose is to monitor a large area and to delimitate the unstable zone, the radar should be installed far from the cliff, using the MATLAB tool to check that the LoS distance remains below 4 or 5 km, depending on the GB-InSAR device. Contrariwise, if the purpose is to characterize the displacement gradient, one will try optimizing the resolution while keeping the LoS as parallel as possible to the

displacement vector. In that case, the tool helps verifying and avoiding the foreshortening and shadowing areas.

Nevertheless, the radar acquisition characteristics are often not the only thing to consider when choosing the best location. Most of the time, the electricity access and an easy installation on a flat surface, as well as the expected instability movement direction, reduce the choices (Caduff et al., 2015).

The tool could be improved and extended to the other GB-InSARs of type ArcSAR or rotary RAR (Pieraccini et Miccinesi 2019) and for the estimation of satellite InSAR images characteristics in order to select the best ascending or descending orbit acquisition before starting the downloading and treatment of the images which can also be a long and laborious work (Berardino et al. 2002; Mancini et al. 2021).

• P9, L158: "A good balance between an acceptable resolution and backscattered enough signal must be found (Figure 6a, b)".You describe that the maximum distance at which the GB-SAR can be deployed is 4-5 km, and that radar parameters have to be adjusted with regard to the distance. How do you choose these parameters in practice? Are you testing different parameters and comparing the results? Or are there specifications in the GB-SAR manual? Have you thought about including in your tool a rough estimate of the backscatter intensity of the point of interest with regard to the radar position, in order to find the appropriate radar parameters at a certain distance. Then the range resolution can be automatically updated with this information.

This could be an improvement for the future indeed and it is a very interesting suggestion. For now, the BW to select is a specification given by the manufacturer and varies depending on the manufacturers and the radar. With Lisalab system, an internal tool permits the selection of the good BW depending on the distance and the number of frequency points. With more frequency points, the acquisition time will take much longer but the BW can be increased for better resolution.

So a good balance between resolution, acquisition time (~frequency points) and BW must be found and this depends also on the monitored surface: In general if you are monitoring a Rockwall or a slope with some boulders in movement the BW must be selected in order that the dimension of the range resolution will be smaller that the dimension of the wedge or of the boulder you want to measure. At a long distance, to keep a High BW and a strong enough backscattered signal, the number of frequency points is increased, increasing also the acquisition time.
If you are measuring an earth landslide since it movement is in general much more homogenous within the unstable area you can chose a smaller BW for decreasing the acquisition time.

But we do not know how this is done with other manufacturer's radar. So we decided to keep it as is and the BW is an input given by the user, since the way the BW is chosen can vary from one manufacturer to the other.

• The structure of the manuscript and also the formatting in certain cases make it difficult to read. Some examples:

◦ P17, L307: The following paragraph is completely redundant, as the necessary input parameters and handling of the tool has already been described in Section 3.2. I do not think this is redundant since the goal of the paragraph is the description of the software interface. The input parameters and output are not described again in this section. But to address your comment, the interface section is shorten and put into the methodology section. We created a new section: "4. **Description of the tested case studies**" for splitting the description of the sites and the results obtained with the tool.

◦ P16, L292: Section 3.3.3 consists of only one sentence, which is not good practice. This was merged with section 3.3.4 (foreshortening and layover maps).

◦ P8, L137: Why is this sentence in bold letters and centered? This was an error and removed

II- English language

I find the manuscript to suffer from poor usage of the English language. Proofreading by a native speaker would improve the reading of the manuscript considerably, I believe. Some examples:

◦ P2, L48: "...consisting in a radar measuring head translating along a rail." → Changed to: moving along the rail

◦ P4, L84: "...the amplitude of the signal sent must be important to reach the Earth surface and to be backscattered with enough intensity to be recorded by the radar receiver" → Changed to: The transmitted signal must have sufficient amplitude to reach the Earth's surface and be backscattered with enough intensity to be detected by the radar receiver

◦ P9, L159: ".. backscattered enough signal.." → Changed to: A good balance between an acceptable resolution and a sufficiently strong backscattered signal must be found

◦ P10, L174: "A good monitoring is when the information related to this SoI is distributed in the maximum of pixels in range and not compressed only in a few ones" → Changed to: A monitoring campaign is effective when information concerning this SoI is distributed across a wide range of pixels rather than compressed within a few; this involve attempting to achieve the finest possible range resolution

◦ P13, L226: "… middle of the area …" → middle is replaced by center

◦ P22, L371: " … the radar image is affected by an important foreshortening …" → important is replaced by strong

III- Minor comments:

• P1, L24: "..it was dedicated". It is still, inter alia, dedicated to studying small movements phenomena. So usage of "has been dedicated" would fit better in my opinion. → This is changed to 'has been dedicated'.

• P4, L62: Section 2.1.1 I think it's confusing why range and azimuth directions are not introduced for GB-SAR here. → This is added :

*In the case of the GB-InSAR, the azimuthal and range directions are parallel and perpendicular to the rail, respectively. dLoS definition is similar to the one in the aerial case but the near-range is the line forming the smaller angle with the horizontal line and the fare-range the larger angle (Figure 1 b).*

• P4, L69: "Those angles" In my opinion, it's not good practice to refer to the paragraph title like that. → The angles are repeated in the text

• P5, L92: "signal emitted is of lower intensity". Compared to what? → It is improved : *is of lower intensity compared to satellite radar emissions*

• P5, L94: Figure 1e and 1g should be 1f and 1h, respectively. → This is corrected.

• P10, L182: The table and its contents could be explained in much more detail, e.g. why the detected displacement is lower than the real displacement.. this can be, by the way, the case in every radar acquisition geometry. → More details are now given in the table.

There is more chance the recorded displacement is less that the real one with P3 if we assume a displacement along the steepest slope. This is clarified in the text:

*LoS may not be parallel to the displacement, the recorded displacement value may be less than the real one, assuming a displacement along the steepest slope.*

• Table 3: Different font sizes used here → This is corrected

• P14, Equation (9): Formatting.. → $\alpha_{app} = \tan^{-1}\left(\tan\alpha \times \sin\left[\omega_{LOS} - \omega_{slope}\right]\right)$      (9)

• P14, L251: " … should encompass the instable area to monitor." Shouldn't be some stable area also be included as reference? → This is changed to : *Should encompass the whole instable area to monitor, as well as an area supposed to be stable for the atmospheric corrections (Pipia et al., 2008; Noferini et al., 2005) and the post-processing unwrapping (Goldstein et al., 1988).*

• P15, Eq. 19, dLos in [Rmin,Rmax] .. does this definitively hold true in case of local topography? Or is the "mean plane" used here? → I do not understand this comment. It is the real dLoS between the radar location and each point of the Point Cloud located within the radar footprint which are considered here.

• P17, L296: What's the "mean plane"? Hasn't been introduced before. → This is clarified in section 3.3.2.

*Once the points illuminated by the radar are known, a mean square method (Wolberg, 2006) is used to determine the mean intersecting plan, defined by its normal vector $\vec{N}$. This vector is then converted into mean slope dip αMEAN and mean slop dip direction ωMEAN.*

• Table 5: dLos of synthetic test should be 200m, I guess. Furthermore, how can dlos in case 3 be outside of the range limits? → It is indeed 200 m.

In case 3, the distance estimated was checked again. There was an error. it is 200m.

• Table 6: i did not get why is there a difference between the dLoS in Table 5 and Table 6? → In table 5 it is a rough estimation of the distance to the monitoring zone while in table 6, it is the real distance between selected radar and target coordinates. In table 5, we changed dLoS to *distance estimation*.

• P26, L411: LoS is usually referred to as a vector or direction, not distance. → I added : "LoS distance"

• Figure A: Colorbar labels of foreshortening and distance maps are mixed up → the label names were mixed up. This is corrected.

• Figures A/B/C: As I understand it, the foreshortening degree should be without unit? Instead of given in degrees.. → It is changed to [-]

• P35, L572: gradientm? Something went wrong in the automatic referencing of the article. This was corrected.

• P36, L623: Where was this published? → The reference is changed (Wolff et al. 2023)

---

## Author Comment (AC2)

Dear reviewer,

Thank you for your thorough review of the article and for providing additional comments, which complement the feedback from the first reviewer. Please find attached our response, which outlines the corrections made to address your insightful comments.

- Section 2. Table 1 presents some problems. Line of sight, in the unit column it would be better to replace "vector" with "unit vector" probably in "m" (a vector has a unit). → We think $\overrightarrow{LoS}$ is a vector because it is defined by a direction (radar to target) and a length. However, we agree that the unit of the vector is m. The corresponding line in the table is changed as follow:

| Name | Abbreviation | Unit | Definition |
|---|---|---|---|
| Line-of-Sight vector | $\overrightarrow{LoS}$ | m | Vector between radar and target points |

  We also noticed that it was cited as $\overrightarrow{dLoS}$ in Figure 9 and in the text. This has been corrected by $\overrightarrow{LoS}$.

- Radar wavelength, the radar is not limited to 0.8 cm - 10 cm (e.g. the L-band is more like 23 cm). → Indeed, the range is wider. To avoid confusion, the sentence about the range is removed since it ranges from 3 cm (~C-bands) to 23 cm (~L-bands) for satellite InSAR, while for the used GB-InSAR, it operates in Ku bands (from 16.7 to 25 mm). The definition is now: Radar wavelength = Spatial period of the signal

- Synthetic antenna length, "L can be infinite", L is constrained by the fact that a target must be during the acquisition within the footprint of the beam and therefore cannot be infinite (even if several km). → The definition is changed to account for the comment:
  - In the case of Linear GB-InSAR, rail length used to focus the radar image (which is shorter than the total rail length). L is generally 2 or 3 m.
  - In the case of satellite InSAR, L can be several km.

- Range Resolution, "vertical" should be "line of sight". → Vertical is changed to "Resolution along the Line of Sight".

- Azimuthal resolution, you should add "parallel to the sensor's motion", "horizontal" does not fully define the direction. → The definition is changed to "Resolution of the radar image along the line parallel to the sensor's motion".

• Section 2.1.4, Table 2 specifies 17.1 to 17.3 GHz, it seems to me that this restriction is only for the Ku band (GBSARs operating on other bands exist), it needs to be clarified. → The restrictions provided in the table are specific to GB-radar operating in the specific BW ranging between 17.1 and 17.3. For clarity, the table is removed, and a better explanation is given directly in the text: Specifically, for GB-InSAR operating in the frequency range of 17.1 to 17.3 GHz, the maximum limits for the frequency bandwidth BW and the power output are 200 MHz and 26 dBm, respectively.

- L260 EQ 11. The equation seems incomplete (at least one vector is missing to the right of the matrix). In addition, an element of the matrix must be missing a "-" sign  (the determinant of the matrix is different from 1 as a rotation matrix should have). → The vector is removed in the left side of the matrix for the homogeneity. We also rechecked the formula and it seems fine. We do not understand where the – should be. We checked also within the code, and by applying this formula, the resulting transformation is ok. We did not change the formula.

However, we agree that there is a problem of consistency in the Equation 13. For the homogeneity of the equation a matrix T defined by the components of $\overrightarrow{LoS}$ is added. The text is now: Each point coordinates can be converted from the *global* geographical coordinate system to the new *local* coordinate system by applying the translation matrix T defined by the vector $\overrightarrow{LoS} \begin{pmatrix} \Delta x_{global} \\ \Delta y_{global} \\ \Delta z_{global} \end{pmatrix}$ followed by the rotation of matrix Ω:

$$\begin{cases} T = \begin{pmatrix} \Delta x_{global} \\ \Delta y_{global} \\ \Delta z_{global} \end{pmatrix} \\ \Omega = \begin{pmatrix} a & -b & 0 \\ b & a & 0 \\ 0 & 0 & 1 \end{pmatrix} \end{cases} \quad .(1)$$

The relation linking the coordinates of each point in the *global* geographical coordinate system and the new *local* coordinate system is:

$$\begin{pmatrix} x \\ y \\ z \end{pmatrix}_{local} = \Omega . \begin{pmatrix} x \\ y \\ z \end{pmatrix}_{global} + T \quad (2)$$

- L306 a typo after "Figure 10" → The typo is removed.
- Table 5: "Location" columns must mention the unit ([m]?) → The unit is added.
- Figure 11 the scale (65) seems too large (compared to the 200m) and does not mention the unit. Maybe just remove (the 200m line could be enough)? → Indeed, there is an error, and the scale is removed from the image as suggested.

---

## Referee Report (RR1)

**Review of egusphere-2023-2489-version3**

I acknowledge that the authors of the manuscript "A new tool for the estimation of Ground-Based InSAR acquisition characteristics before starting installation and monitoring survey" addressed my comments and concerns. I still have some comments regarding the updated version:

- If the comparison of satellite-borne InSAR and GB-InSAR is one of the two main goals of the manuscript, it could be helpful to add this aspect somehow to the title of the manuscript.

- Fig. 1: For me, the chirp looks like a linear frequency modulated chirp, but below, you plot the frequency spectrum of a stepped chirp.

- Equation 3: The real pulse length of the chirp in pulsed systems is not the inverse of the bandwidth! This is what I was already stating in my first review. The effective pulse length after pulse compression with a matched filter is the inverse of the bandwidth.

- Paragraphs 4.1.1 and 4.1.2.: Align paragraph titles.

- Table 2/4/5, Figure 7, Equation 12/13, etc.: Formatting.. Adjust font sizes.